# Japan’s Dental Care Facing Population Aging: How Universal Coverage Responds to the Changing Needs of the Elderly

**DOI:** 10.3390/ijerph18179359

**Published:** 2021-09-04

**Authors:** Etsuji Okamoto

**Affiliations:** Department of Health & Welfare Management, University of Fukuchiyama, Kyoto 620-0886, Japan; okamoto-etsuji@fukuchiyama.ac.jp

**Keywords:** universal health coverage, health insurance claims, administrative data, claims database

## Abstract

Although universal health coverage (UHC) is pursued by many countries, not all countries with UHC include dental care in their benefits. Japan, with its long-held tradition of UHC, covers dental care as an essential benefit, and the majority of dental care services are provided to all patients with minimal copayment. Being under UHC, the scope of services as well as prices are regulated by the uniform fee schedule, and dentists submit claims according to the uniform format and fee schedule. The author analyzed the publicly available dental health insurance claims data as well as a sampling survey on dental hygiene to illustrate how Japan’s dental care is responding to the challenges from population aging. A marked improvement was found in dental health status in the elderly population as measured by improved tooth-specific survival. The improvement may be attributable to the universal coverage of dental care, as evidenced by the steady increase in home visits by dentists/dental hygienists as well as home oral rehabilitation services.

## 1. Introduction

Oral health is indispensable for healthy aging because good nutrition cannot be sustained without healthy eating and digestion. This is particularly true for Japan, which boasts the world’s longest longevity (81.41 years for males and 87.45 years for females in 2019). To ensure healthy aging, dental health status must be improved in proportion to the prolonged lifespan [1]. However, monitoring dental health on a population level may not be feasible in all countries. Japan has advantages, in that (1) it has universal health coverage including dental care, and it is feasible to grasp the utilization of dental care on a national level using health insurance claims data; and (2) it has conducted a national sampling survey to monitor the dental health of the entire nation periodically (Dental Hygiene Survey, DHS).

The research question of this article is two-fold: (1) how has dental health status improved despite population aging, and (2) how has the utilization of dental care services changed to accommodate the changing needs of the rapidly aging population?

In most countries, dental care is not covered by public insurance, or the coverage, if any, may not be universal. On the other hand, dental care is covered by Japan’s universal health insurance, with some exceptions (e.g., orthodontics). Under Japan’s universal health insurance, the prices of each procedure as well as of medicines are regulated by government as a form of national uniform fee schedule. Because of such generous coverage, it is technically feasible to grasp utilization of dental care services through national statistics. Dentists submit itemized claims every calendar month for each patient. The submitted claims data are stored in the national claims database (NDB), and the itemized statistics from a month (typically every May) are published as the “Social Insurance Claims Survey (SICS)”. Further, summary statistics of NDB also became available as “NDB open data (NDBOD)” after 2014. By combining NDB open data and the SICS, one can illustrate the utilization of dental care services.

## 2. Materials and Methods

In this study, the author relied on two kinds of data on dental health: a sampling survey on dental health status evaluated by dentists (the Dental Hygiene Survey, DHS) and administrative data of health insurance claims submitted by all dental offices. The latter is particularly unique to Japan’s dental care, because Japan has a universal health coverage and all dental care services are covered under a uniform fee schedule as well as a uniform claims format. Almost all insurance claims are submitted electronically and are accumulated in a large database known as the National Claims Database (NDB), from which aggregate data are publicly available either as (1) the Social Insurance Claims Survey (SICS) or (2) the NDB open data (NDBOD).

The time frame was inevitably limited by the data availability. Although the DHS and the SICS has been conducted since 1957, it is only recently that these data have been publicly available in Excel format (DHS was provided only after the 2006 survey, and SICS became a population survey only after 2012; NDBOD became available only after 2014, and dental data were not included or very limited in the initial few years).

The raw data used in tables and figures are provided as a Appendix A in Excel format.

### 2.1. Dental Hygiene Survey

To survey the dental health status of the entire population, the DHS is conducted by the Ministry of Health, Labour and Welfare (MHLW) as a sampling survey at an interval of five to six years as part of the National Health and Nutrition Survey (the DHS had been conducted at the interval of six years since 1957, but the interval was shortened to five years after the 10th survey in 2011). The author analyzed tooth-specific and age-specific survival, comparing the latest 2016 results and 2005 results [2].

The sample was selected to reflect the dental health status of the entire population. A total of 150 National Census districts were selected, and all residents were subjects of the survey. However, the sex and age distribution of the sample may not properly reflect Japan’s population structure, as shown in Table 1 and Table 2.

### 2.2. National Claims Database Open Data (NDBOD)

The National Claims Database (NDB) is arguably one the largest administrative databases in the world and started to accumulate the data of medical, dental, and pharmaceutical claims in 2009 [3]. The dental data became publicly available as “open data” in 2014, and the data are aggregated by clinical diagnoses broken down by sex and five-year age groups. One limitation of the data is that it only provides the number of diagnoses, and one claim may contain more than one diagnoses. Another limitation of the NDB is a legal one: due to the strict privacy protection rule, data smaller than 10 are omitted (*one-digit suppression rule*). Therefore, one should be cautioned that the data provided by the NDBOD may underestimate the real figures.

### 2.3. Social Insurance Claims Survey (SICS)

The SICS is another survey of claims data derived from the NDB. The differences from the NDBOD are that (1) SICS contains monthly data (typically for May in the survey year), while the NDBOD provides annual data, and (2) the one-digit suppression rule does not apply. SICS contains data on the number of claims, the number of office visits, and the monetary values for every clinical procedure.

## 3. Results

Analysis was conducted to answer the two research questions presented in the introduction section. Dental health status was evaluated using the DHS to illustrate how dental health status changed over the survey period, and the change in the utilization of dental care services was evaluated by health insurance claims data using SICS and NDBOD.

### 3.1. Change of Dental Health Status

#### 3.1.1. Survival of Teeth of the Elderly

Survival of teeth of the elderly (65 years or older) has improved even during such a short period of 11 years. As shown in Figure 1, age-specific survival of permanent teeth has improved markedly between the 2005 and 2016 surveys. The improvement is more prominent in the older age groups. Japan Dental Association launched the “80-20 campaign” in 1989, which means “maintaining at least 20 teeth at the age of 80” [4]. According to the Dental Hygiene Survey in 1999, 80-year-old persons had an average of eight teeth remaining, and only 15% of them had 20 teeth or more remaining. According to the Dental Hygiene Survey in 2016, 51.2% of the elderly at age 80 had 20 teeth or more remaining.

#### 3.1.2. Tooth-Specific Survival of the Elderly

The DHS surveys for each tooth. The following table [Table 3] illustrates the tooth-specific improvement in survival for the elderly (≥65-year-old). The most improvement in survival was found in the left lower second molar, which showed 1.54-fold improvement in survival (48.4% survival in 2016 as opposed to 31.5% in 2005). On the other hand, the least improvement was in the right lower canine, which showed 1.16-fold improvement in survival over the 11-year interval. Lower canines have the highest survival rates of all teeth (82.5% for right and 84.1% for left remaining in the elderly in DHS 2016), and the improvement was inevitably limited.

#### 3.1.3. Conditions of Teeth of the Elderly

Although the survival of teeth showed a marked improvement over the 11-year interval, the conditions of teeth of the elderly (≥65-year-old) did not show much difference over the same period. It is remarkable that the percent of complete dentures among the missing teeth declined from 50.7% in 2005 to 39.7% in 2016. Also remarkable was that the percent of implants increased from 0.3% of missing teeth in 2005 to 1.3% in 2016 Table 4.

### 3.2. Analysis of Diagnoses Contained in Health Insurance Claims

Health insurance claims contain diagnoses, which reflect the changing pattern of utilization of dental care services by different age groups. NDBOD provides diagnoses and was used for analysis. However, NDBOD does not include the data on the number of claims, and the data from another source (the Medical Care Benefit Survey, MCBS) was used as a supplement to calculate the number of diagnoses per claim. The MCBS is another survey on health insurance claims, providing only aggregate data on the number of claims or health care expenditures.

#### *N* of Diagnoses by Diagnostic Categories

The number of diagnoses contained in dental claims stored in NDB has increased steadily. However, one should be cautioned that the NDB stores only electronically submitted claims, and the computerization of claims was not well-developed in the early years. In addition, the latest 2018 data include approximately 213 million dental claims. There is an increasing trend in the number of diagnoses contained in a claim. The number of diagnoses per claim increased from 1.32 diagnoses per claim in 2014 to 1.63 in 2018 (Table 5).

The following graph shows a declining share of dental caries with aging, possibly reflecting the declining number of remaining teeth. On the other hand, the share of periodontal diseases remains constant over aging. However, one should be cautioned in interpreting the NDB data. According to the “one-digit suppression (numbers less than 10 will not be displayed)”, the number of claims may be substantially underestimated (Figure 2).

When broken down by ICD10-level diagnoses, two diagnoses (periodontitis and dental caries) account for 57.2% of the total number, and the top ten diagnoses account for 82.8% of the total diagnoses (Table 6).

### 3.3. Analysis of the Change in Utilization of Dental Care Services 

The SICS has been conducted every year since 1957. It was conducted as a sampling survey when health insurance claims were submitted in paper form. Since 2012, it started to extract data from NDB and became a population survey instead of a sampling survey.

The SICS data covers only a one-month period (May in the survey year) and is therefore affected by seasonal variation. In addition, one should be reminded that the latest data in 2020 are severely affected by the COVID19 epidemic.

The author focuses on home care because it reflects the population aging. As shown in Table 7, home care is provided mainly to the elderly population. The rapid proliferation of home visits/home care management provided by dental offices is illustrated by the % increase between 2015 and 2019. In only four years, the utilization of home visits increased by 37.9% and home care management by 51.1%. The age distribution of home care services provision shows a sharp contrast to that of initial office visits, indicating that home care services are provided predominantly to the disabled elderly at home.

#### 3.3.1. Dental Home Visits

Dental home visits are provided to patients who cannot visit dental clinics due to physical handicaps, and the number of dental home visits is increasing steadily (Table 8). For the elderly who are living in nursing homes or long-term care facilities, dentists can visit more than one patient at a time. Considering the time saving in such cases, the fee for dental home visits is set considerably lower for multiple patients in a building [11,000 yen for one patient and 3610 yen for the second or more patients in a building. The fee is further reduced to 1850 yen for ten or more patients in a building].

#### 3.3.2. Dental Home Care Management

Dental home care management (DHCM) is a surcharge to home visit fees. While home visit fees are reimbursed on every visit, dental home care management is considered to be a professional service by dentists, involving a planned, scheduled and long-term management of patients to maintain their oral health and nutrition. Therefore, DHCM is reimbursed once a month, and while home visits may be provided by any dental practitioners, dental home care management is expected to be provided by specially designated dental clinics called “home care supporting dental clinics” (HCSDC).

There are certain conditions for dental clinics to be designated HCSDCs. To qualify as HCSDC type 1, the clinic must provide 15 times or more home visits per year, and for type 2, 10 times or more.

In addition to the requirement for the number of home visits, the following conditions must be met [5]:must have at least one dentist who completed a training course on geriatric dentistry as well as risk management for emergencies;must have at least one dental hygienist.provide patients and/or family members information on home visits in writing;must be affiliated with other HCSDCs for back-up;must have provided at least five home visits in response to requests from long-term care facilities (nursing homes, care managers, visiting nursing stations, etc.).

As shown in Figure 3, the number of DHCMs has been increasing. Since 2018, HCSDCs have been divided into two categories: type 1 and 2. Type 1 HCSDCs are entitled to a higher management fee (3200 yen per month per patient) than type 2 (2500 yen) because they must meet the more stringent conditions than type 2 HCSDCs. There is no difference between the two types as to the function and role of the dental clinics, and the distinction is intended to provide economic incentives by way of a differential level of financial reimbursement to encourage more clinics to contribute to home care for the elderly.

#### 3.3.3. Home Visits by Dental Hygienists

A study analyzing the data on dental clinics and patients’ behavior demonstrated the importance of dental hygienists in influencing the patients’ behavior [6] and tooth loss [4]. Home visits by dental hygienists are also covered by health insurance [Table 9]. Conditions for reimbursement include: (1) dental hygienists must spend at least 20 min per visit, and (2) reimbursement is capped at four times per month.

There was a major revision in the fee schedule for dental hygienists in 2018. Until then, the fee for dental hygienists fell into two categories: simple (1200 yen) and complicated (3000 yen). However, from 2018 onward, the fee schedule was revised to the same structure used with dentists: 3600 yen for the 1st patient in a building, 3280 yen for the 2nd to 9th patient in a building, and 3000 yen for the 10th patient or more.

#### 3.3.4. Home Oral Rehabilitation Services

Home oral rehabilitation was added to health insurance benefits in 2016 as a surcharge to dental home visits. The fee is reimbursed when dentists provide oral rehabilitation services to the patients who are charged for “dental home visits”. The conditions for reimbursement are (1) patients must have an eating disorder requiring a constant dental management, (2) dentists must develop a long-term dental management plan, and (3) dentists spend at least 20 min on site [5].

Since this fee is a surcharge to dental home visits, the fee is categorized by the number of teeth under management, and not by the number of patients in a building [Table 10].

## 4. Discussion

Oral health of the elderly population in Japan has improved considerably, as evidenced by the Dental Hygiene Surveys. The share of the elderly who maintain 20 or more teeth has increased from 15% in 1999 to 51.2% in 2016 [7]. Tooth-specific survival has improved by 32% over the eleven-year interval. A non-systemic review of dental health of the elderly concluded that “the epidemiological literature on oral health in the elderly is not very encouraging, and it indicates profound imbalances among countries and regions” [8]. If so, Japan may be viewed as one of the few successful countries [9]. However, it is difficult to identify factors contributing to the improvement of dental health status. Improved dental care targeting the middle-aged population with periodontal diseases must have been a major factor [10], but such measures alone will not accommodate the ever-increasing number of elderly patients who cannot visit dental clinics without assistance.

Japan is one of the rare examples of countries that not only cover dental care in their universal health insurance system but expand the coverage to home dental care to accommodate elderly patients at home. Since Japan has universal health coverage and dental care has been included in the benefit, it was possible to illustrate the utilization of dental services as well as the number of diagnoses contained in a claim, particularly after the full computerization was achieved and a national database accumulating the claims data was established. Previous studies analyzing home dental care services relied on questionnaire surveys of practicing dentists in limited areas [11,12,13]. The present study has methodological advantages over preceding ones in that it used health insurance claims data, a more accurate and reliable data source than questionnaires, to illustrate the proliferation of home dental care services. Health insurance claims contain not only procedures but also diagnoses. This enables researchers to illustrate how the prevalence of diseases has shifted through the population’s aging. Reflecting the aging of the entire population, the prevalence of major diagnostic categories has shifted gradually, with an increasing share for periodontal diseases, while the share of missing teeth has decreased due to the improved survival of teeth of the elderly.

Japan’s uniform fee schedule is revised every two years and serves as a policy implementation tool for the government [14]. The government has increased the coverage of home care services in both medical and dental care. Utilization of home care services has increased steadily in recent years and is expected to increase further, reflecting the aging population (there was a sharp drop in the year 2020; this reflects the impact of the COVID19 epidemic and may be viewed as a temporary phenomenon).

The author analyzed publicly available data source including sampling surveys on dental status as well as health insurance claims data. However, one should be cautioned concerning the limitations and drawbacks of the data. Although Japan’s claims data are uniform and comprehensive, they lack information on the socio-economic status of patients. For example, one recent study revealed “a pro-rich inequality for both income and wealth” in dental care use under universal public coverage [15]. This is a research question the present study failed to answer.

Monitoring the socio-economic disparity in oral health status would be the future challenge for researchers.

## 5. Conclusions

Japan has covered dental care as benefit of its universal health coverage. Thanks to such generous coverage, people can receive dental care with a minimal copayment. The dental health status as measured by survival of teeth has improved considerably, particularly for the elderly population. In response to the rapidly aging population, an increasing trend of home dental care services for the elderly patients has been observed. Although the overall performance of Japan’s dental care for the elderly has been satisfactory, some matters such as socio-economic disparity remain unaddressed and leave room for future research.

## Figures and Tables

**Figure 1 ijerph-18-09359-f001:**
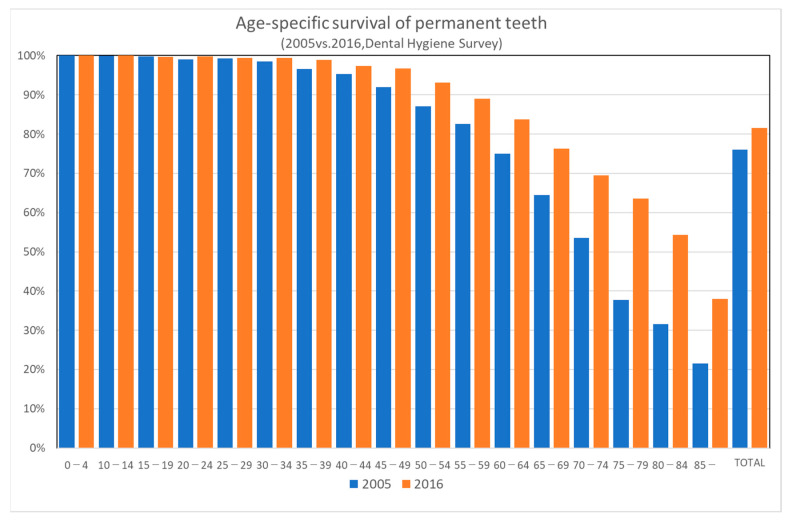
Age-specific survival of permanent teeth.

**Figure 2 ijerph-18-09359-f002:**
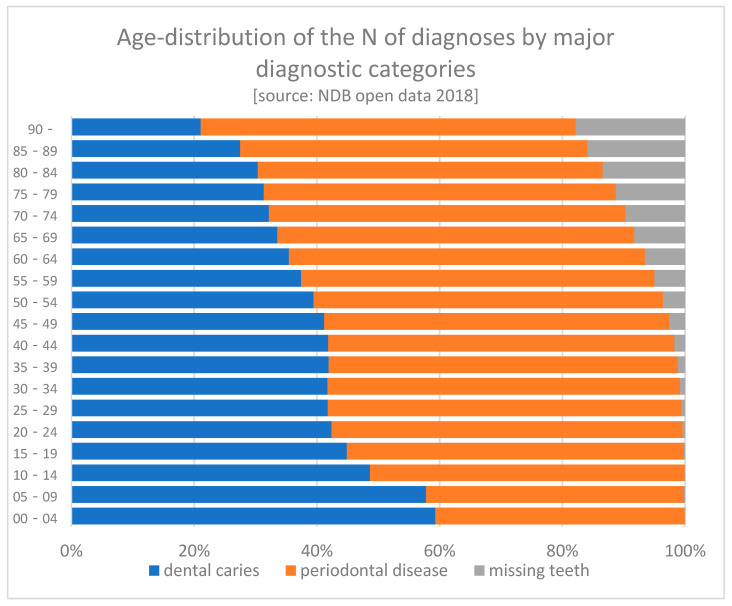
Age distribution of the number of diagnoses by major diagnostic categories.

**Figure 3 ijerph-18-09359-f003:**
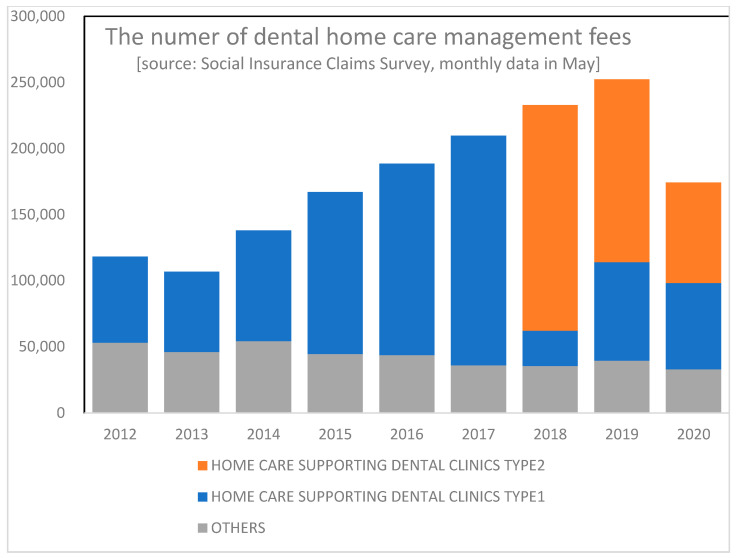
The number of dental home care management fees.

**Table 1 ijerph-18-09359-t001:** The number of subjects of the Dental Hygiene Survey.

*N* of Subjects (Dental Hygiene Survey)						
		2005			2016	
	F	M	MF	F	M	MF
05~09	130	117	247	94	100	194
10~14	92	116	208	58	64	122
15~19	65	54	119	32	19	51
20~24	58	47	105	36	34	70
25~29	103	71	174	49	37	86
30~34	142	97	239	95	44	139
35~39	139	58	197	124	66	190
40~44	173	74	247	157	97	254
45~49	164	95	259	125	77	202
50~54	192	105	297	140	81	221
55~59	249	158	407	154	100	254
60~64	242	192	434	213	138	351
65~69	288	208	496	258	245	503
70~74	227	221	448	196	184	380
75~79	183	138	321	164	155	319
80~84	104	67	171	125	99	224
85~	46	26	72	72	64	136
TOTAL	2597	1844	4441	2092	1604	3696

**Table 2 ijerph-18-09359-t002:** Age distribution of the sample of the Dental Hygiene Survey vs. population.

Dental Hygiene Survey (2016)		Population Pyramid (2015)
85~	136	3,117,257
80~84	224	4,961,420
75~79	319	6,276,856
70~74	380	7,695,811
65~69	503	9,643,867
60~64	351	80,455,010
55~59	254	7,515,246
50~54	221	7,930,296
45~49	202	8,662,804
40~44	254	9,732,218
35~39	190	8,316,157
30~34	139	7,290,878
25~29	86	6,409,612
20~24	70	5,968,127
15~19	51	6,008,388
10~14	122	5,599,317
05~09	194	5,299,787

**Table 3 ijerph-18-09359-t003:** Tooth-specific survival rate of the elderly and improvement over 11 years (2005–16).

Tooth-Specific Survival Rate of the Elderly (≥65) and Improvement over 11 Years (2005–16)									
				2005 (*N* = 1508)			2016 (*N* = 1562)		2016/2005
			Present [P]	Missing [M]	P/(P + M)	Present [P]	Missing [M]	P/(P + M)	
lower	R	median incisor	890	617	0.59	1173	387	75.2%	1.27
		lateral incisor	940	563	0.63	1215	341	78.1%	1.25
		canine	1068	438	0.71	1288	274	82.5%	1.16
		first premolar	872	636	0.58	1148	414	73.5%	1.27
		second premolar	723	785	0.48	983	576	63.1%	1.32
		first molar	508	999	0.34	781	781	50.0%	1.48
		second molar	518	987	0.34	741	820	47.5%	1.38
		third molar	148			187			
	L	median incisor	895	613	0.59	1188	372	76.2%	1.28
		lateral incisor	955	551	0.63	1225	333	78.6%	1.24
		canine	1070	436	0.71	1314	248	84.1%	1.18
		first premolar	884	624	0.59	1182	378	75.8%	1.29
		second premolar	718	789	0.48	968	594	62.0%	1.30
		first molar	494	1014	0.33	746	816	47.8%	1.46
		second molar	473	1030	0.31	755	806	48.4%	1.54
		third molar	136			168			
upper	R	median incisor	760	748	0.50	1059	502	67.8%	1.35
		lateral incisor	773	732	0.51	1062	495	68.2%	1.33
		canine	868	640	0.58	1158	402	74.2%	1.29
		first premolar	712	795	0.47	1008	550	64.7%	1.37
		second premolar	662	845	0.44	964	597	61.8%	1.41
		first molar	596	912	0.40	861	701	55.1%	1.39
		second molar	561	946	0.37	817	744	52.3%	1.41
		third molar	73			82			
	L	median incisor	728	780	0.48	1048	514	67.1%	1.39
		lateral incisor	733	774	0.49	1051	509	67.4%	1.39
		canine	843	664	0.56	1139	422	73.0%	1.30
		first premolar	716	792	0.47	988	571	63.4%	1.33
		second premolar	667	841	0.44	919	642	58.9%	1.33
		first molar	638	870	0.42	881	681	56.4%	1.33
		second molar	540	968	0.36	805	754	51.6%	1.44
		third molar	67			75			
			21,229	21,389	0	28,979	15,224	65.6%	1.32

※ all differences between 2005 and 2016 are statistically significant at *p* = 0.01.

**Table 4 ijerph-18-09359-t004:** Status of teeth of the elderly.

Status of Teeth of the Elderly (≥65, Dental Hygiene Survey)					
	2005 (*N* = 1508)		2016 (*N* = 1562)		
present teeth	21,229	(100%)	28,979	(100%)	
sound teeth	8035	(37.8%)	11,924	(41.1%)	**
with dental sealant	2	(0.0%)	5	(0.0%)	ns
colored	366	(1.7%)	325	(1.1%)	**
not colored	7667	(36.1%)	11,594	(40.0%)	**
filled teeth					
Crown, not bridge abutment	2411	(11.4%)	2677	(9.2%)	**
Crown, bridge abutment	6143	(28.9%)	6861	(23.7%)	**
Root cap			58	(0.2%)	NA
filling	3165	(14.9%)	6101	(21.1%)	**
decayed teeth					
Caries incipient	774	(3.6%)	750	(2.6%)	**
Caries high grade	701	(3.3%)	608	(2.1%)	**
missing teeth	21,389	(100%)	15,253	(100%)	
implant	62	(0.3%)	192	(1.3%)	**
bridges	1394	(6.5%)	1585	(10.4%)	**
no prosthesis	2341	(10.9%)	2646	(17.3%)	**
complete denture	10,849	(50.7%)	6060	(39.7%)	**
partial denture	6743	(31.5%)	4741	(31.1%)	ns
removal for orthodontics	0	(0.0%)	29	(0.2%)	**
	42,618		44,232		

ns: not significant, **: significant at *p* = 0.01.

**Table 5 ijerph-18-09359-t005:** *N* of dental claims by major diagnostic categories.

*N* of Dental Claims by Major Diagnostic Categories					
	Dental Caries	Periodontal Disease	Missing Teeth	Total *N* of Diagnoses	*N* of Claims (*) *N* of Diagnoses/Claim
2014	107,549,905	140,143,615	16,389,285	264,082,805	200,612,846
	40.7%	53.1%	6.2%	100%	1.32
2015	128,939,230	169,719,402	19,330,067	317,988,699	204,865,945
	40.5%	53.4%	6.1%	100%	1.55
2016	130,172,602	174,024,015	19,215,128	323,411,745	210,679,509
	40.2%	53.8%	5.9%	100%	1.54
2017	132,611,007	184,154,163	19,024,386	335,789,556	212,878,244
	39.5%	54.8%	5.7%	100%	1.58
2018	134,161,234	193,685,587	18,840,620	346,687,441	212,916,550
	38.7%	55.9%	5.4%	100%	1.63

Source: NDB open data, (* *N* of claims: Medical Care Benefit Survey).

**Table 6 ijerph-18-09359-t006:** Ten most common diagnoses of dental claims.

Ten Most Common Diagnoses of Dental Claims			
	*N* of Diagnoses	% Diagnoses	Cumulative %
periodontitis	135,915,272	39.2%	39.2%
dental caries	62,368,855	18.0%	57.2%
gingivitis	15,891,362	4.6%	61.8%
missing teeth	13,615,550	3.9%	65.7%
pulpitis	13,253,313	3.8%	69.5%
chronic periodontitis	12,653,681	3.6%	73.2%
apical periodontitis	12,303,356	3.5%	76.7%
dental caries of 2nd degree	7,958,861	2.3%	79.0%
dental caries treated	7,357,028	2.1%	81.1%
acute purulent periodontitis	5,868,951	1.7%	82.8%
· · ·	· · ·		
TOTAL	346,687,441	100%	100%

Source: NDB open data 2018.

**Table 7 ijerph-18-09359-t007:** Age distribution of home care/visits and initial office visits (2015 and 2019).

Age Distribution of Home Care Management/Visits and Initial Office Visits (2015&19)									
	Dental Home Care Management			Dental Home Visits			Initial Office Visits		
	2015	2019	2019/15	2015	2019	2019/15	2015	2019	2019/15
00–04	60	78	130.0%	121	250	206.6%	245,093	235,314	96.0%
05–09	106	155	146.2%	201	379	188.6%	549,344	518,384	94.4%
10–14	115	174	151.3%	251	348	138.6%	319,308	313,197	98.1%
15–19	228	301	132.0%	553	620	112.1%	152,345	149,726	98.3%
20–24	574	786	136.9%	1268	1743	137.5%	186,978	193,072	103.3%
25–29	807	1033	128.0%	1934	2130	110.1%	247,407	238,322	96.3%
30–34	997	1420	142.4%	2301	2989	129.9%	299,769	280,451	93.6%
35–39	1576	1747	110.9%	3761	3781	100.5%	358,134	327,538	91.5%
40–44	2167	2753	127.0%	5566	6362	114.3%	408,947	388,877	95.1%
45–49	2222	3719	167.4%	5839	8952	153.3%	362,846	428,651	118.1%
50–54	2407	3849	159.9%	6814	9703	142.4%	347,693	388,702	111.8%
55–59	2795	4209	150.6%	8698	11,887	136.7%	345,556	370,018	107.1%
60–64	4259	5316	124.8%	15,003	16,325	108.8%	407,955	381,468	93.5%
65–69	6704	8450	126.0%	27,775	30,838	111.0%	471,160	465,311	98.8%
70–74	10,890	13,690	125.7%	48,519	53,301	109.9%	417,259	460,625	110.4%
75–79	18,528	25,711	138.8%	85,367	106,754	125.1%	323,419	398,915	123.3%
80–84	32,005	43,829	136.9%	157,923	190,088	120.4%	217,166	258,361	119.0%
85–89	38,508	60,312	156.6%	196,367	278,578	141.9%	101,746	131,748	129.5%
90–	41,996	74,662	177.8%	189,865	320,775	168.9%	33,003	48,203	146.1%
total	166,944	252,194	151.1%	758,126	1,045,803	137.9%	5,795,128	5,976,883	103.1%

Source: Social Insurance Claims Survey.

**Table 8 ijerph-18-09359-t008:** The number of dental home visits.

The Number of Dental Home Visits				
	One Patient in a Building	2~9 Patients in a Building	10 or More Patients in a Building	Total
2012	205,646	568,505		774,151
2013	175,332	430,464		605,796
2014	228,198	277,838	421,356	927,392
2015	245,716	356,415	544,678	1,146,809
2016	270,419	382,967	597,183	1,250,569
2017	305,904	430,549	648,807	1,385,260
2018	361,446	628,846	615,724	1,606,016
2019	370,991	667,835	642,843	1,681,669
2020	295,306	467,573	447,467	1,210,346

Source: Social Insurance Claims Survey.

**Table 9 ijerph-18-09359-t009:** The number of home visits by dental hygienists.

The Number of Home Visits by Dental Hygienists						
	Simple	Complicated	One Patient in a Building	2~9 Patients in a Building	10 or More Patients in a Building	Total
2012	82,621	224,294				306,915
2013	77,292	190,608				267,900
2014	101,178	225,424				326,602
2015	99,210	270,097				369,307
2016	107,608	299,310				406,918
2017	115,556	333,746				449,302
2018			16,961	54,203	379,964	451,128
2019			17,263	58,139	415,659	491,061
2020			11,165	35,785	282,732	329,682

Source: Social Insurance Claims Survey.

**Table 10 ijerph-18-09359-t010:** The number of home oral rehabilitation services.

The Number of Home Oral Rehabilitation Services				
	<10 Teeth	10~20 Teeth	20≤ Teeth	Total
2016	1931	952	1441	4324
2017	2696	1608	2392	6696
2018	4798	2541	3931	11,270
2019	6352	3645	5686	15,683
2020	4623	2891	4408	11,922

Source: Social Insurance Claims Survey.

## Data Availability

Dental Hygiene Survey: 2005 Survey—https://www.e-stat.go.jp/stat-search/file-download?statInfId=000031411439&fileKind=0; https://www.e-stat.go.jp/stat-search/file-download?statInfId=000031411440&fileKind=0; [accessed 2 September 2021]. 2016 Survey—https://www.e-stat.go.jp/stat-search/file-download?statInfId=000031607230&fileKind=0. [accessed 2 September 2021]. NDB open data: 2014 data: https://www.mhlw.go.jp/file/06-Seisakujouhou-12400000-Hokenkyoku/0000139460.xlsx [accessed 2 September 2021]; 2015 data: https://www.mhlw.go.jp/file/06-Seisakujouhou-12400000-Hokenkyoku/0000177285.xlsx [accessed 2 September 2021]; 2016 data: https://www.mhlw.go.jp/content/12400000/000347784.xlsx [accessed 2 September 2021]; 2017 data: https://www.mhlw.go.jp/content/12400000/000711946.xlsx [accessed 2 September 2021]; 2018 data: https://www.mhlw.go.jp/content/12400000/000539792.xlsx. Social Insurance Claims Survey: https://www.e-stat.go.jp/stat-search/files?page=1&toukei=00450048&tstat=000001029602 [accessed 2 September 2021].

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
