# Peer review of "Japan’s Dental Care Facing Population Aging: How Universal Coverage Responds to the Changing Needs of the Elderly"

_ijerph, 2021, doi:10.3390/ijerph18179359_

Round 1

Reviewer 1 Report

The paper just presents selected data about the dental health care in Japan - even without clear identification of sources (critical formal mistake). It does not include any research question (or even real research goal), methodology and literature review are missing. Is information it can have some value, but  from the point of criteria for real academic paper, its value is very low.

Author Response

[point1] The paper just presents selected data about the dental health care in Japan - even without clear identification of sources (critical formal mistake).

[response1] If my definition of the source data was insufficient, please accept my apology.   I added the following description as an introduction of the “Materials and Methods” section and hope it will suffice.

In this study, the author relied on two kinds of data on dental health: a sampling survey on dental health status evaluated by dentists (the Dental Hygiene Survey, DHS) and administrative data of health insurance claims submitted by all dental offices.   The latter is particularly unique to Japan’s dental care because Japan has a universal health coverage and all dental care services are covered under a uniform fee schedule as well as a uniform claims format.   Almost all insurance claims are submitted electronically and are accumulated in a large database known as the National Database (NDB), from which aggregate data are publicly available either as 1) Social Insurance Claims Survey (SICS) or the NDB open data.

[point2] It does not include any research question (or even real research goal), methodology and literature review are missing. Is information it can have some value, but from the point of criteria for real academic paper, its value is very low.

[response2] I corroborated the research question by adding the following explanation at the beginning of the introduction section.   I admit that this paper is by no means a hypothesis-testing one but do believe it is of value in that it demonstrates the contributory factor of the improved dental health to the prolonged lifespan using the big data of dental care, which is not readily available in other countries.

Oral health is indispensable for healthy aging because good nutrition cannot be sustained without healthy eating and digestion.   This is particularly true for Japan, with its world-longest longevity (81.41 years for male and 87.45 years for females in 2019).   To ensure healthy aging, the dental health status must be improved in proportion to the prolonged lifespan.   However, monitoring the dental health on a population level may not be feasible in all countries.   Japan has advantages in that 1) it has universal health coverage including dental care and it is feasible to grasp the utilization of dental care on a national level using health insurance claims data and 2) it has conducted a national sampling survey to monitor the dental health of the entire nation periodically (Dental Hygiene Survey, DHS).

The research question of this article is two-fold: 1) to analyze how the dental health status has improved despite population aging and 2) how the utilization of dental care services changed to accommodate the changing need of the rapidly aging population.

Reviewer 2 Report

This is an interesting paper that provides important information about dental care in Japan and demonstrates trends in dental care and dental health.  A few things need to be addressed, as per my following comments:

1.  "To survey the dental status of the entire population, the Dental Hygiene Survey (DHS) is conducted as a sampling survey at the interval of five to six years as part of the National Health & Nutrition Survey. The author analyzed tooth-specific and age-spe- cific survival comparing the latest 2016 results and 2005 results."    Why were those dates chosen (esp. 2005)?

2.  It is unclear to me what the Table 1 column “N of Subjects” is referring to since the numbers of subjects under M and F headings is vastly different.  Should that be “Age of Subjects”?

3.  "Japan Dental Association launched the “80- 20 campaign” in 1989, which means “maintaining at least 20 teeth at the age of 80” [iii]. According to the Dental Hygiene Survey in 1999, 80 year old persons had an average of eight teeth remaining and only 15% of them had 20 teeth or more remained. According to the Dental Hygiene Survey in 2016, 51.2% of the elderly aged 80 year old had 20 teeth or more remained."   Is this a statistically significant difference?

4.  "The DHS surveys for each tooth. The following table illustrates the tooth-spe- cific improvement in survival for the elderly (>=65 year old). The most improvement in survival was found in the left lower second molar, which showed 1.54 fold improvement in survival (48.4% survivial in 2016 as opposed to 31.5% in 2005). On the other hand, the least improvement was right lower canine, which showed 1.16 fold improvement in survival over 11 years interval. Lower canines have the highest survivals in all teeth (82.5% for right and 84.1% for left remaining in the elderly in DHS 2016) and the improvement was inevitably limited."  Is this statistically significant?  This paper would greatly  benefit from some inferential statistical analysis to determine how statistically significant are the findings.  

5.  Table 3 and Table 4 would benefit from inferential statistical analysis to demonstrate whether findings from one year to the next are significant.

6.  "The number of diagnoses contained in dental claims stored in NDB has increased steadily. However, one should be cautioned that the NDB stores only electronically submitted claims and the computerization of claims was not well developed in the early years. Also, the latest 2018 data include approximately213 million dental claims. There is an increasing trend in the number of diagnoses contained in a claim. The number of diagnoses per claim increased from 1.32 diagnoses per claim in 2014 to 1.63 in 2018."  Again, is this significant?

7.  Table 5:  Change “detanl” to “dental”.

8.  Need to define HCSDCs type 1 and type 2 better.

9.  In the Discussion:  "The improvement is attributable to the increased health insurance coverage to home dental care services, which predominantly are consumed by the elderly patients."  Correlation does not necessarily beget causation.  What other factors might explain dental improvement?

Author Response

[point1].  "To survey the dental status of the entire population, the Dental Hygiene Survey (DHS) is conducted as a sampling survey at the interval of five to six years as part of the National Health & Nutrition Survey. The author analyzed tooth-specific and age-specific survival comparing the latest 2016 results and 2005 results." Why were those dates chosen (esp. 2005)?

[response1] The data were chosen conveniently because of availability.   The author added the following paragraph in the methodology section.

   The time frame was inevitably limited by the data availability.   Although, the DHS and the SICS has been conducted since 1957, it is only recently when these data are publicly available as Excel format (DHS is provided only after the 2006 survey and SICS became a population survey only after 2012.   NDB open data became available only after 2014 and dental data were not including or very limited in the initial few years).

[point2] It is unclear to me what the Table 1 column “N of Subjects” is referring to since the numbers of subjects under M and F headings is vastly different. Should that be “Age of Subjects”?

[response2] The N of subjects refers to the number of people selected for dental examinations.   Since the DHS is a sampling survey, it is necessary to describe the sample size in an age-specific manner.   MF, of course, refers to sex and obviously more females were represented in the sample.   Table2 compares the sample of DHS 2016 and the population pyramid showing that they are by no means comparable.

[point3] "Japan Dental Association launched the “80- 20 campaign” in 1989, which means “maintaining at least 20 teeth at the age of 80” [iii]. According to the Dental Hygiene Survey in 1999, 80 year old persons had an average of eight teeth remaining and only 15% of them had 20 teeth or more remained. According to the Dental Hygiene Survey in 2016, 51.2% of the elderly aged 80 year old had 20 teeth or more remained."   Is this a statistically significant difference?

[response3] The sample size of the 2016 survey 80 years or over was 360.   The sample size of the 1999 survey was 6903 but no age-specific breakdown was given [MHLW https://www.mhlw.go.jp/topics/0105/tp0524-1.html].   However, it would be safe to assume that the 1999 survey also had the same sample size for the population 80 years old or over (=360).   In 1999, given the point estimate of 15%, the 95%C.I. would be calculated as 11.31%~18.69%.   The sheer rate of 51.2% in 2016 is far beyond the 95%CI in 1999.  One can confidently conclude that the rate of 51.2% of elderly over 80 with 20 or more teeth is significantly higher than the rate of 15% in 1999 beyond chance.

[point4] "The DHS surveys for each tooth. The following table illustrates the tooth-specific improvement in survival for the elderly (>=65 year old). The most improvement in survival was found in the left lower second molar, which showed 1.54 fold improvement in survival (48.4% survivial in 2016 as opposed to 31.5% in 2005). On the other hand, the least improvement was right lower canine, which showed 1.16 fold improvement in survival over 11 years interval.  Lower canines have the highest survivals in all teeth (82.5% for right and 84.1% for left remaining in the elderly in DHS 2016) and the improvement was inevitably limited."  Is this statistically significant?  This paper would greatly benefit from some inferential statistical analysis to determine how statistically significant are the findings.

[response4] As discussed in [response3], even the lower canines showed a significant improvement survival in 2016 at p=0.01.   Although the difference was smaller than other teeth, the improvement of canines defies chance.

[point5] Table 3 and Table 4 would benefit from inferential statistical analysis to demonstrate whether findings from one year to the next are significant.

[response6] I calculated the 99%CI for all survival rate of the data in 2005 in table 3 and found that all rate of 2016 were significantly higher than 2005 at p=0.01.   I revised table 3 by adding “All differences are statistically significant at p=0.01”

As for table 4, all differences were significant at p=0.01 but with some exceptions.   Table4 was revised accordingly.

[point6]"The number of diagnoses contained in dental claims stored in NDB has increased steadily. However, one should be cautioned that the NDB stores only electronically submitted claims and the computerization of claims was not well developed in the early years. Also, the latest 2018 data include approximately 213 million dental claims. There is an increasing trend in the number of diagnoses contained in a claim. The number of diagnoses per claim increased from 1.32 diagnoses per claim in 2014 to 1.63 in 2018."  Again, is this significant?

[response6] I am afraid that the point6 on statistical significance is NOT applicable to NDB because it is a population survey and not a sampling one.   Given the enormous size of the data (213 million), the observed increase of 1.63 diagnoses per claim in 2018 over the 1.32 in 2014 defies chance.

[point7] Table 5:  Change “detanl” to “dental”.

[response 7] Thank you for pointing out my silly mistake.

[point 8] Need to define HCSDCs type 1 and type 2 better.

[response8] The author added the following passage for clarification of the purpose of classifying into two categories.  There is no difference between the two types as to the function and role of the dental clinics and the distinction is intended to provide economic incentives by way of differential level of financial reimbursement to encourage more clinics to contribute to home care to the elderly.

[point9] In the Discussion: "The improvement is attributable to the increased health insurance coverage to home dental care services, which predominantly are consumed by the elderly patients." Correlation does not necessarily beget causation. What other factors might explain dental improvement?

[response9]

The author supplemented the discussion by adding the following passage as well as a citation.

A non-systemic review of dental health of the elderly concluded that “the epidemiological literature on oral health in the elderly is not very encouraging, and it indicates profound imbalances among countries and regions”[[i]].   If so, Japan may be viewed as one of the few successful countries.   However, it is difficult to identify factors contributing to the improvement of dental health status.   Improved dental care targeting middle-aged population with periodontal diseases must have been a major factor [[ii]], but such measure alone will not accommodate the ever-increasing number of elderly patients who cannot visit dental clinics without assistance.   Japan is one of the rare examples of countries which not only cover dental care in its universal health insurance system but expand the coverage to home dental care to accommodate elderly patients at home.

[i] Gil-Montoya JA, et al. Oral health in the elderly patients and its impact on general well-being: a nonsystematic review. Clinical Interventions in Aging. 2015:10:461-467.

[ii] Furuta M, et al. Periodontal status and self-reported systemic health of periodontal patients regularly visiting dental clinics in the 8020 Promotion Foundation Study of Japanese Dental Patients. J of Oral Science, 61(2):238-245 (2019).

Reviewer 3 Report

Authors with focus on the aged-population in Japan evaluated the dental health, hygiene and insurance claims of Japan's dental care under universal health coverage. By utilizing data extracted from publicly available domains, authors reported their research findings. Manuscript presents interesting results.

COMMENTS:

  1. The bulk of the manuscript lacks references to several statements within the body of text, thereby making it appear as author opinion statements or general phenomena
  2. There is no summary statement of result findings/conclusion in the abstract section. Authors need to provide at least a sentence highlighting their results in this section
  3. Authors adopted year 2005 and 2016 for most of the analysis. It will be interesting to know how the age-specific survival of permanent teeth occurred in year 2010/2011, since survey is done every 5-6 years as alluded in material section. Was there an uptick in elderly people's teeth number within this time frame also, when compared to year 2005?
  4. For Table 7, why was 2019 selected? why not compare trend from 2015-2019? The methods section of manuscript gives no indication of which year should be reviewed. Consequently, re-define research objective focus and clearly indicate in methods section also, so as to have a designated time frame for the various parameters being measured.
  5. Why use roman numbers for the few references cited in manuscript?

Author Response

[point1] The bulk of the manuscript lacks references to several statements within the body of text, thereby making it appear as author opinion statements or general phenomena.

[response1] The author supplemented references to strengthen the argument so that the article will not be a mere statement of personal views or opinions.

[point2] There is no summary statement of result findings/conclusion in the abstract section. Authors need to provide at least a sentence highlighting their results in this section

[response2] The author supplemented the abstract by adding the following concluding remark following the main text.

A marked improvement was found in dental health status in the elderly population as measured by improved tooth-specific survival.   The improvement may be attributable to the universal coverage of dental care as evidenced by the steady increase of home-visits by dentists/dental hygienists as well as home oral rehabilitation services.

[point3] Authors adopted year 2005 and 2016 for most of the analysis. It will be interesting to know how the age-specific survival of permanent teeth occurred in year 2010/2011, since survey is done every 5-6 years as alluded in material section. Was there an uptick in elderly people's teeth number within this time frame also, when compared to year 2005?

[response3] The DHS is a cross-sectional survey and it is not possible to compare with previous years such as 2005/6 or 2010/11.   The DHS has been conducted every six years since 1957 and the interval was shortened to five years after the 10th (2011) survey.   Although, there is a total of 11 survey results including the latest 2016 survey, only the last three results (survey 2005, 2011 and 2016) are provided as Excel files over the government statistics portal site (www.e-stat.go.jp).   Since the author wanted to compare as long time span as possible, two surveys (2005 and 2016) were conveniently chosen.   The data had to be reformatted to be handled by pivot-table.   The reformatting process was time-consuming and the author had to disregard the 2011 data and there is no telling if there was a “uptick” in elderly’s teeth number.   Still, the author believes that it would be safe to say each survey results are statistically sound enough to represent the dental health of the entire population and comparison of 2005 and 2016 will provide a reasonable estimate of the change of dental health of 11 years interval even without the middle data.   Also, the author added the following clarification to the description of the DHS.  (The DHS has been conducted at the interval of six years since 1957 but the interval was shortened to five years after the 10th survey in 2011)

[point4] For Table 7, why was 2019 selected? why not compare trend from 2015-2019? The methods section of manuscript gives no indication of which year should be reviewed. Consequently, re-define research objective focus and clearly indicate in methods section also, so as to have a designated time frame for the various parameters being measured.

[response4] I modified the Table7 as suggested to compare the trend between 2015 and 19.   Also, I found a data extraction error in dental home visits and revised accordingly (error was not serious enough to change findings or conclusions).   I appreciate the reviewer and apologize for the error.

The author added the following paragraphs in methodology and results section respectively.

Methodology section

   The time frame was inevitably limited by the data availability.   Although, the DHS and the SICS has been conducted since 1957, it is only recently when these data are publicly available as Excel format (DHS is provided only after the 2006 survey and SICS became a population survey only after 2012.   NDB open data became available only after 2014 and dental data were not including or very limited in the initial few years).

Results section

The rapid proliferation of home visits/home care management provided by dental offices is illustrated by the % increase between 2015 and 2019.   In only four years, the utilization of home visits increased by 37.9% and home care management by 51.1%.   The age distribution of home care services provision shows a sharp contrast to that of initial office visits indicating that home care services are provided predominantly to the disabled elderly at home.

[point5] Why use roman numbers for the few references cited in manuscript?

[respone5] I changed them to Arabic numerals.

Round 2

Reviewer 1 Report

The author now formulated two research questions as follows:

1) to analyze how the dental health status has improved despite population aging, and

2) how the utilization of dental care services changed to accommodate the changing need of the rapidly aging population.

One step forward, but not enough. The analytical part should be organized according to these RQs and not only according to the source. The text should also better show own contribution of the author. What is the value added? The fact that dental health improved is now somehow confirmed (may be by too many data from existing databases), however the try to speak about success factors is not based on any analysis – I would avoid any discussion in this direction. Conclusions are poor. Tables are not referenced.

Author Response

[point] One step forward, but not enough. The analytical part should be organized according to these RQs and not only according to the source. The text should also better show own contribution of the author. What is the value added? The fact that dental health improved is now somehow confirmed (may be by too many data from existing databases), however the try to speak about success factors is not based on any analysis – I would avoid any discussion in this direction. Conclusions are poor. Tables are not referenced.

[response] The results section was reorganized according to the research questions, not the data source.   The added value of this study resides in its methodological strength relying on the health insurance claims data, more accurate and comprehensive than questionnaire surveys.   The author supplemented three citations analyzing on the development of home dental care, all of which relied on questionnaire surveys.   Although the conclusions of the present study may not differ much from preceding studies, the present study provides richer and more generalizable conclusions than preceding studies (one should note that all preceding studies surveyed in limited areas only while the data used in this study covers the entire country.   For example, two the three studies were in Tokyo and another in Iwate prefecture only.).   As for table 5 and 10, data sources were added because they were missing.

Reviewer 2 Report

No additional comments for authors.  My questions have been answered.

Author Response

[reviewer]

No additional comments for authors.  My questions have been answered.

[author] I appreciate the reviewer for valuable suggestions and hope they improved the quality of my manuscript.

Reviewer 3 Report

Authors have made material changes to the earlier submitted manuscript. However, the referencing and citation of referred materials is insufficiently done, as evident by the total number of references

Author Response

Authors have made material changes to the earlier submitted manuscript. However, the referencing and citation of referred materials is insufficiently done, as evident by the total number of references

[response] I enriched the discussion section by supplementing references.   The methodological strength of the present study was elucidated by contrasting with three preceding studies which solely relied on questionnaire surveys.
